# Satellite Imaging-Based Residential Greenness and Accelerometry Measured Physical Activity at Midlife—Population-Based Northern Finland Birth Cohort 1966 Study

**DOI:** 10.3390/ijerph17249202

**Published:** 2020-12-09

**Authors:** Soile Puhakka, Tiina Lankila, Riitta Pyky, Mikko Kärmeniemi, Maisa Niemelä, Katja Kangas, Jarmo Rusanen, Maarit Kangas, Simo Näyhä, Raija Korpelainen

**Affiliations:** 1Department of Sports and Exercise Medicine, Oulu Deaconess Institute Foundation sr, PO Box 365, 90100 Oulu, Finland; tiina.lankila@oulu.fi (T.L.); riitta.pyky@odl.fi (R.P.); mikko.karmeniemi@oulu.fi (M.K.); katja.kangas@odl.fi (K.K.); maarit.kangas@oulu.fi (M.K.); raija.korpelainen@odl.fi (R.K.); 2Center for Life Course Health Research, Faculty of Medicine, University of Oulu, PO Box 5000, 90014 Oulu, Finland; 3The Geography Research Unit, Faculty of Science, University of Oulu, PO Box 3000, 90014 Oulu, Finland; jarmo.rusanen@oulu.fi; 4Medical Research Center Oulu, Oulu University Hospital and University of Oulu, PO Box 5000, 90014 Oulu, Finland; 5Research Unit of Medical Imaging, Physics, and Technology, Faculty of Medicine, University of Oulu, PO Box 5000, 90014 Oulu, Finland; maisa.niemela@oulu.fi; 6Natural Resources Institute Finland, Economics and Society, University of Oulu, PO Box 413, 90014 Oulu, Finland; 7Center for Environmental and Respiratory Health Research, University of Oulu, PO Box 5000, 90014 Oulu, Finland; simo.nayha@oulu.fi

**Keywords:** green space, GIS, cohort study, adults, linear regression, generalized additive model

## Abstract

Background: Recently, the importance of light physical activity (LPA) for health has been emphasized, and residential greenness has been positively linked to the level of LPA and a variety of positive health outcomes. However, people spend less time in green environments because of urbanization and modern sedentary leisure activities. Aims: In this population-based study, we investigated the association between objectively measured residential greenness and accelerometry measured physical activity (PA), with a special interest in LPA and gender differences. Methods: The study was based on the Northern Finland Birth Cohort 1966 (5433 members). Participants filled in a postal questionnaire and underwent clinical examinations and wore a continuous measurement of PA with wrist-worn Polar Active Activity Monitor accelerometers for two weeks. The volume of PA (metabolic equivalent of task or MET) was used to describe the participant’s total daily activity (light: 2–3.49 MET; moderate: 3.5–4.99 MET; vigorous: 5–7.99 MET; very vigorous: ≥8 MET). A geographic information system (GIS) was used to assess the features of each individual’s residential environment. The normalized difference vegetation index (NDVI) was used for the objective quantification of residential greenness. Multiple linear regression and a generalized additive model (GAM) were used to analyze the association between residential greenness and the amount of PA at different intensity levels. Results: Residential greenness (NDVI) was independently associated with LPA (unadjusted *β* = 174; CI = 140, 209) and moderate physical activity (MPA) (unadjusted *β* = 75; CI = 48, 101). In the adjusted model, residential greenness was positively and significantly associated with LPA (adjusted *β* = 70; CI = 26, 114). In men, residential greenness was positively and significantly associated with LPA (unadjusted *β* = 224; CI = 173, 275), MPA (unadjusted *β* = 75; CI = 48, 101), and moderate to vigorous physical activity (MVPA) (unadjusted *β* = 89; CI = 25, 152). In women, residential greenness was positively related to LPA (unadjusted *β* = 142; CI = 96, 188) and inversely associated with MPA (unadjusted *β* = −22; CI = −36, −8), vigorous/very vigorous physical activity (VPA/VVPA) (unadjusted *β* = −49; CI = −84, −14), and MVPA (unadjusted *β* = −71; CI = −113, −29). In the final adjusted models, residential greenness was significantly associated only with the amount of LPA in men (adjusted *β* = 140; CI = 75, 204). Conclusions: Residential greenness was positively associated with LPA in both genders, but the association remained significant after adjustments only in men. Residential greenness may provide a supportive environment for promoting LPA.

## 1. Introduction

Although previous studies on the association between physical activity (PA) and health have mostly focused on moderate to vigorous physical activity (MVPA), in longitudinal studies on accelerometry measured PA—including daily habitual activities such as casual walks, gardening, household chores, and slow cycling—light physical activity (LPA) has been positively associated with health [1,2]. The importance of LPA has been considered in new health recommendations on PA [3]. The techniques for measuring objective LPA have taken a step forward [4], and there is growing epidemiological evidence that frequent bouts of LPA can have several positive health outcomes. For example, LPA can improve cardiovascular health, especially among those whose risk factors are already high (e.g., type 2 diabetes) [5], as well as in adults generally [6]. In addition, both longitudinal and cross-sectional studies on adults reported that LPA reduces the risk of mortality [7,8,9]. Still, physical inactivity is a global pandemic, and new approaches are needed to encourage people to be more physically active.

Exposure to greenness has many beneficial health outcomes. Green environments have been shown to be positively associated with the amount of PA, especially LPA, in adults [10,11,12]. As greenness encourages PA, it provides several health benefits. Green environments have been suggested to offer healthier physiological and psychological premises for PA because of less noise and cleaner air [13,14]. The psychological advantage of spending time in green environments is mainly based on the increased capability of directing attention in such surroundings and the presence of vegetation and water, which provide positive signals that indicate a safe and aesthetic environment [15,16] that appeals to our senses [17]. All these features calm our autonomic nervous system and improve restoration [18,19,20]. PA in green environments is likely to provide positive experiences, which may encourage people to visit these places frequently [21,22]. Compared to PA indoors, PA in a green environment is more motivating, requires less exertion [23,24,25], and is more restorative [26]. As green surroundings can provide a more aesthetic and encouraging environment for walking and cycling, PA in such environments is longer lasting and more regular [10,27,28].

Despite some evidence of the positive association between residential greenness and PA, how much greenness in a residential area is really needed to support PA is still unclear. This kind of information, produced by satellite imaging could be valuable for land use development, especially when designing urban areas, which often lack green spaces. While there exists some evidence on nature’s positive effect on PA, comprehensive studies based on large population data with the device-based measurement of PA are lacking.

The aim of this population-based study was to determine the association between residential greenness and PA at midlife, including gender differences, with special emphasis on low-intensity PA. Our hypothesis was that residential greenness is positively associated with the amount of all intensity levels of PA.

## 2. Materials and Methods

The Northern Finland Birth Cohort 1966 study (NFBC1966) includes all those in Northern Finland whose expected dates of birth fell in 1966 and who were invited to the study (*n* = 12,058 live births) [29]. They have been followed up regularly since their birth. This cross-sectional study analyzed the data obtained from the most recent time point, at 46 years of age (*n* = 10,321). The data was collected (from 2012 to 2014) through health-care records, questionnaires (*n* = 6384), and clinical examinations (*n* = 5852). Their PA was measured by wrist-worn Polar Active accelerometers for 14 days (*n* = 5481). The final study population included all those whose residential environmental features were objectively measured by a geographic information system (GIS) (*n* = 5433). The study was approved by the ethical committee of the Northern Ostrobothnia Hospital District in Oulu, Finland (94/2011), and it was performed in accordance with the Declaration of Helsinki. The subjects and their parents provided written consent for the study. The personal identity information was encrypted and replaced with identification codes to provide full anonymity.

### 2.1. Accelerometer-Measured Physical Activity

PA was measured with an accelerometry-based wrist-worn waterproof activity monitor, the Polar Active (Polar Electro Oy, Kempele, Finland). The monitor was blinded, giving no feedback to the user. The Polar Active provides metabolic equivalent (MET) values every 30 s [30] and has been shown to correlate (*r* = 0.86) with the double-labeled water technique, assessing energy expenditure during exercise training [31]. It uses the height, weight, gender, and age of the user as predefined inputs. The monitors were given to the participants during clinical examinations, with the participants being instructed to mail them back after the measurement period. The participants were asked to wear the Polar Active monitors 24 h per day for at least 14 days, including while sleeping, on the wrist of the nondominant hand. The daily averages of time spent in different activity levels (very light: 1–1.99 MET; light: 2–3.49 MET; moderate: 3.5–4.99 MET; vigorous: 5–7.99 MET; very vigorous: ≥8 MET) were calculated for all the participants [32]. Wear time during waking hours (min/day) was calculated as the sum of all five activity levels. The “very light” activity level describes sedentary behavior and thus was not analyzed in this study. The first day, when the activity monitor was given to the participant, was excluded from the analysis. Participants with four or more eligible days (wear time of at least 600 min/day) were included in the study [33,34].

The PA of the participants was expressed as the total daily average MET-minutes spent in LPA, moderate PA (MPA), vigorous/very vigorous (VPA/VVPA), and MVPA. Finally, each PA intensity level was used separately as a dependent variable to analyze the associations between residential greenness and different intensities of PA. The volume of each PA intensity level in MET-minutes was calculated by multiplying each MET value with its duration (30 s).

### 2.2. Geographical Information System

GIS—a tool for gathering, analyzing, and mapping data based on location—was used to assess the quantitative features of each individual’s residential environment. Spatial information on the participant’s immediate residential environment was based on the exact geographical coordinates of the participant’s home during 2014. A circular buffer with a 1 km radius was fixed around each participant’s residency to represent the everyday living environment. Similar-sized buffers have been used in several other studies [35]. Only periods of residence that had lasted at least 3 months were included [36]. In the case of several periods of residence ≥3 months during 2014, the annual average of the normalized difference vegetation index (NDVI) was used. The participants were located all around Finland, but a fifth of them lived in Northern Finland (the data collection in 2014), and 5% of them lived in Helsinki, the metropolitan area of Finland. ArcGIS Pro 2.1 (Redlands, CA, USA) was used to calculate the environmental variables [37].

### 2.3. Normalized Difference Vegetation Index

Of the many tools for measuring greenness, the NDVI has been widely used in both epidemiological and environmental studies [38,39] and has proven to be a reliable tool for measuring residential greenness [40]. In this study, it was used to assess the surrounding greenness within a 1 km buffer of each participant’s residential environment. This method is based on satellite imaging (resolution 30 × 30 m) and provides quantitative information on the land cover’s greenness [41]. Healthy green vegetation (chlorophyll) reflects infrared and green light, and it can also absorb red and blue light. The NDVI uses the following formula: *NDVI = (NIR − R)/(NIR + R)*,(1)
in which the *NDVI* is calculated from the red (*R*) and near infrared (*NIR*) values. Values of the *NDVI* range from −1 to +1. Values close to −1 indicate water bodies, rock, and snow. Values close to 0 (such as 0.2–0.3), in turn, indicate densely built surfaces or other surfaces with sparse vegetation. Highly positive values (>0.6) indicate areas with very dense and healthy green vegetation, such as forests and paddocks [41]. The NDVI was measured from Landsat 8 (L8) satellite images administrated by the USGS (United States Geological Survey) [42]. Images with less than 10% cloud cover were selected, and the months of June to July (2013–2016) were used in the calculation as they represent the greenest months in Finland’s seasonal variation.

### 2.4. Body Mass Index

The weight and height of the participants were measured in the clinical examination, and their body mass index (BMI) was calculated as weight (kg) divided by height squared (m^2^).

### 2.5. Urban–Rural Classification

The 2010 urban–rural spatial classification of Finland is a GIS-based area classification provided by the Finnish Environment Institute. The data was calculated with overlaying 250 × 250 m grid cells, and the classification is based on several statistical features such as population, standard industrial classification of the workforce [43], CORINE Land Cover, commuting, potential accessibility, and area density of buildings. The classification consists of two main regional classes, (1) urban areas and (2) rural areas, which are further divided into seven regional classes: (1) inner urban area, (2) outer urban area, (3) peri-urban area, (4) rural area close to urban areas, (5) local center in rural areas, (6) rural heartland area, and (7) sparsely populated area. In this study, classes 1–3 were combined (1 = “urban areas”), and classes 4–7 were grouped together (2 = “semirural and rural areas”) [44].

### 2.6. Number of Sports Facilities

The number of sports facilities within each participant’s residential environment (1 km circular buffer) was calculated based on the data contributed by the University of Jyväskylä (Lipas Sports Facility GIS Database 2014). This data includes information on public sports facilities in Finland, mainly the municipal ones, as well as a number of sites run by private companies or associations [45].

### 2.7. Questionnaire

The participants completed a postal questionnaire concerning their demographic features, perceived health, health behavior, and socioeconomic background (SES). They rated their perceived health (1 = “Good”; 2 = “Pretty good”; 3 = “Moderate”; 4 = “Pretty poor”; 5 = “Poor”), estimated their life satisfaction (1 = “Very satisfied”; 2 = “Somewhat satisfied”; 3 = “Somewhat dissatisfied”; 4 = “Very dissatisfied”; 5 = “I do not know”), and answered questions concerning their smoking habits (1 = “7 days a week”; 2 = “5–6 days a week”; 3 = “2–4 days a week”; 4 = “Once a week”; 5 = “Occasionally”; 6 = “I do not smoke”) and daily alcohol consumption (g/day). The participants estimated their alcohol consumption with given examples, which were converted to grams. Additionally, information on marital status (1 = “Married”; 2 = “Common-law marriage”; 3 = “Single”; 4 = “Divorced”; 5 = “Divorced from a registered partnership”; 6 = “Widow”; 7 = “Widowed after a registered partnership”), number of children under 18 years old, level of education (1 = “Nonvocational”; 2 = “Professional course”; 3 = “Vocational”; 4 = “Formal post-secondary level”; 5 = “Polytechnic degree”; 6 = “University degree”; 7 = “Other education [describe]”; 8 = “Incomplete education”), and annual household income (€) was acquired. Physically strenuous work was also evaluated (“To what extent are the following tasks and postures part of your job?”). The participants had to evaluate certain tasks and postures in their work: “Heavy physical work in which the body has to struggle”; “Lifting loads of 1–15 kg”; “Lifting loads over 15 kg”; “Continuous movement or walking”; “Repetitious work movements”; “Standing”; “Working with the upper arms elevated”; “Forward-bent work postures”; and “Rotational movements of the trunk.” The response scale was from 1 to 5: 1 (not at all/very rarely), 2 (rarely), 3 (moderately), 4 (often), and 5 (very often). The scale was reclassified as physically light work (light work, 1–2) and strenuous work (strenuous work, 3–5). We summed up the recoded answers of the nine questions and used the variable as continuous [46].

### 2.8. Statistical Methods

The study variables are described in terms of means, standard deviations (SD), and ranges. The statistical significance of the differences in the total amount of PA among groups was analyzed using an independent-samples *t*-test. Spearman’s correlation test was used to test the correlation between PA and the continuous variables. The association among residential greenness (NDVI), other determinants, and PA (at each intensity level and for men and women separately) was analyzed with multiple linear regression analysis. The explanatory variables were tested for multicollinearity, and thus, the highest two-tailed correlation allowed was 0.6. The variance inflation factor (VIF) value was supposed to remain below 10 and the tolerance measure above 20. In the second step, influential observations were handled using Cook’s distance (<3) and excluded from the analyses. The missing data was excluded through the pairwise deletion of cases. All the variables were then forced into the model. The residuals were checked for normal distribution, and the results are presented as betas (β) (unadjusted beta and adjusted beta) together with their confidence intervals (95% CI). The highest limit for the statistical significance of the models was set at *p* < 0.05 and was estimated with ANOVA. The data was analyzed using the PASW Statistics software [47].

In addition, a generalized additive model (GAM) was used, as a secondary method, to study the association between residential greenness (NDVI) and PA at each intensity level. In the GAM, the linearity of the effects is not required [48]. The GAM permits both linear and nonlinear response shapes as well as a combination of them within the same model [49] and can be used to find curvilinearity and potential threshold values for the variable of interest [50]. Natural cubic splines with four degrees of freedom were fitted using the ns function available in the R software, version 4.01 (https://www.r-project.org/). The normality of the residuals was checked. The restricted maximum likelihood (REML) was used in the GAM.

## 3. Results

The characteristics of the study population are presented in Table 1.

### 3.1. Physical Activity

The amount of PA at different intensity levels according to socioeconomic factors, lifestyle, and residential environment is presented in Table 2. Women had more LPA than men, though men had more MPA, VPA/VVPA, and MVPA compared to women. Participants with low levels of education were more likely to have more LPA and MPA than highly educated participants. Participants with high levels of education had higher amounts of VPA. Participants without children under 18 years old had more LPA, and participants with children under 18 years old had more MPA. Nonsmokers had more MPA, VPA, and MPVA than current smokers. Participants who reported having better perceived health also had more LPA, MPA, VPA/VVPA, and MVPA.

### 3.2. Factors Associated with Physical Activity

Table 3 summarizes the associations of greenness with various levels of PA in terms of crude and adjusted regression coefficients. In LPA, a one-unit increase in the NDVI was associated with an increase of 174 MET-minutes (95% CI 140, 209), but adjusting for personal and environmental factors decreased this effect to 70 MET-minutes (CI 26, 114). The crude and adjusted effects on LPA were more pronounced in men (224 MET-min [CI 173, 275] and 140 MET-min [CI 75, 204], respectively). In women, only the crude coefficient indicated the effect of greenness on LPA (142 MET-min [CI 96, 188]), which, however, reduced to insignificance after adjustments.

At intensity levels higher than LPA, no overall increase of PA was observed, although in men, the crude coefficient for MPA was 75 MET-minutes (CI 48, 101) and that for MVPA was 89 MET-minutes (CI 25, 152). However, at higher intensity levels, the PA of women decreased with increasing greenness; MPA decreased by 12 MET-minutes (CI 7, −23), VPA/VVPA decreased by 36 MET-minutes (CI 13, −86), and MVPA decreased by 48 MET-minutes (CI 10, −108) per unit change in the NDVI.

Appendix ATable A1 and Table A2 show the regression coefficients for all variables used in the adjusted models. In addition to the NDVI, which was the variable of interest in this study, various levels of PA were associated with other factors, although in most cases, their effects were smaller than that of the NDVI. Factors that increased LPA included having children younger than 18 years, the female gender, high education, physically strenuous work, and a low number of sports facilities in the area. Good self-rated health was the strongest factor underlying VPA/VVPA and MVPA in both men and women.

The shape of the association between residential greenness and LPA is shown in Figure 1 in terms of smoothed estimates based on the GAM. The association was positive and almost linear, with no threshold value. In the unadjusted analysis, the effect of greenness ranged from −130 MET-minutes at the lowest NDVI values to 66 MET-minutes at the highest NDVI values, with a respective range of −72 MET-minutes to 15 MET-minutes in the adjusted analysis.

## 4. Discussion

The purpose of this study was to investigate the relationship between satellite imaging-based residential greenness and the amount of accelerometry measured PA at different intensities, with a special emphasis on LPA and gender differences. We found a significant positive association between residential greenness and the total daily average volume of MET-minutes of LPA and MPA. Residential greenness was significantly associated with both men’s and women’s LPA. In the adjusted models, a positive connection was found only between residential greenness and LPA. When studying the association with men and women in the adjusted models, we discovered a positive connection only between residential environment and men’s total daily average volume of MET-minutes of LPA. We found no threshold values between residential greenness and all the intensity levels of PA, indicating that even a small increase in greenness may result in increased PA levels.

Previous studies support our findings on the association between residential greenness and LPA. It is becoming increasingly evident that greenness provides suitable environments for low-intensity outdoor exercise types such as walking, slow cycling, activities with animals (dog walking, leisure riding), and gardening. Previously, a positive connection between greenness and walking among middle-aged women was suggested [39], and in another study, greenness was related not only to initiating recreational walking but also, more importantly, to maintaining recreational walking in adults [28]. The daily amount of LPA may result from commuting as well, which is important with regard to daily PA. However, LPA is often performed outdoors and is encouraged by factors such as well-being, the effects of nature and aesthetic settings, better air quality, and less noise [13,14]. Greenness has been shown to have restorative effects, which can be highly motivating for working-age people. Pottering around outdoors, such as through gardening and habitual chores, can be a pleasant hobby without performance pressures but with the secondary benefit of increased PA [21].

However, given the lack of longitudinal studies, we still cannot determine whether the green environment caused those people to be active or whether they were just generally more active individuals. According to Shanahan’s theory [21], the benefits of green exposure and PA can be divided into three stages: subadditive, synergistic, and additive. This theory can give us support to better understand the interaction between greenness and PA. The synergistic effect can occur only when PA is performed while exposed to a green environment. For example, we may find exercising more refreshing only outdoors because the surrounding factors are simply better (fresh air, less noise, etc.). Greenness can support PA by adding some advantages to it. For example, it can encourage PA to be performed more regularly or be longer lasting. [51] Finally, a subadditive approach is less simple but possible. According to this approach, the benefits of PA can limit our ability to benefit from nature [21]. Even if the environment is pleasant, PA performed above our own comfort zone can decrease the positive experience of nature. However, these interactions remain uncertain, and future studies should focus on them to better understand them.

Other factors could explain the connection between residential greenness and LPA, which requires deeper inspection into subjective aspects and especially socioeconomic background. For instance, people whose residential environment is very green, especially in the countryside, have less possibilities for indoor exercising because of the lack of sports facilities and the long distances between homes and services. In addition, choosing a green residential environment as a place to live could be a subjective matter as well; people who enjoy outdoor activities may prefer living in greener residential environments. In this case, greenness motivates them to spend time outdoors, making them physically active as well. Such motivation is driven by the need to experience the aesthetic view, especially green environments [24,52]. However, areas with lower socioeconomic status are often located in rural areas or outside city centers [53]. In Finland, affordable housing expenses might be the major factor for people who live in those areas, not necessarily residential greenness.

We observed that the positive association between residential greenness and LPA was stronger among men than women. Finding an explanation for this is not straightforward and requires further studies. According to the Working Paper of the Finnish Forest Research Institute [54] women spend more time outdoors within their residential areas such as their own yards, parks, and natural environments.

The positive linear association between residential greenness and LPA confirms that land use design should focus on developing and bringing greenness to areas such as those with densely built urban landscapes. There are many ways to increase residential greenness. Particularly, big and densely built cities need green spaces that people can easily exploit for their daily purposes. Many studies on well-being and PA have drawn attention to green parks, which are important spaces especially in bigger cities, where green spaces are often limited [55,56]. Parks are ideal spaces for LPA, and adult urban citizens have been suggested to use parks for sports more than adult rural citizens [57]. In addition, previous studies suggest that green parks are positively associated with leisure walking among middle-aged people [58]. This is particularly noteworthy because parks are often the only green environments that cities can offer.

The role of residential greenness in the daily LPA of citizens should be considered when finding new approaches to encourage PA. We also propose that instead of only MVPA, further studies should pay more attention to LPA. The observation that regular, longer-lasting LPA is good for our health is becoming more evident. Previous studies have proposed that the daily volume of PA is more important than the intensity of the exercise, yet the evidence applies mostly to young adults [59,60]. Our finding on lower BMI and its positive association with LPA supports the notion that LPA, not just MPVA, can help improve our body composition. In addition, increased LPA can be easily adapted into daily routines and serve the needs of different groups of people with varied backgrounds. Additionally, new tools and approaches to observe people’s daily movements in residential environments need to be further studied.

### Strengths and Weaknesses

Our study was based on a large population-based sample of 46-year-old men and women located all around Finland. PA and residential greenness were accelerometry measured, and all the known uncertainties concerning the methods were taken into consideration. However, some limitations in the study were observed. Due to the cross-sectional study setting, we cannot assert causality between greenness and PA. Participating in a follow-up study (in this case, on PA) can cause unusual behavior, such as temporarily increased PA that is not a part of the individual’s normal behavior. In addition, LPA can be partly a result of work-related PA, and the wrist-worn activity monitor used to assess PA accelerometry could not accurately detect bicycling. Even though the methodology can be used in various population and areas, the results of this study may not be generalized to different populations, age groups, or regions outside Northern Europe.

The calculation of the NDVI is inevitably sensitive to topographic and atmospheric factors, although this was addressed via careful data selection and processing. Water bodies were not eliminated from the GIS data, which may have led to lower NDVI values in buffers with higher quantities of water bodies. Additionally, by using buffers, we did not have data on the participants’ perception of greenness, which should be considered in further studies. Buffers are artificially created, and their capability to show the participants’ real-life movements is limited. However, without GPS-based information on each individual’s movements, their usability is moderately reliable given the focus on objective greenness and PA from a phenomenal perspective in our study. Self-selection bias concerning moving to green areas as a part of subjective behavior may exist.

## 5. Conclusions

Studies on residential greenness and PA suggest that greenness near homes can promote PA, especially LPA. Our study confirmed that residential greenness was positively connected to LPA, especially for men. We also found that a small increase in greenness may have a significant effect on LPA. PA combined with the green environment’s health benefits may lead to positive health outcomes in general. The knowledge gained in this study can be used in interventions that aim to promote PA. Nature-based interventions are cost-effective and easily implemented. However, longitudinal studies and new methods to study people’s daily movements and exposure to greenness are vital in deepening the knowledge in this field.

## Figures and Tables

**Figure 1 ijerph-17-09202-f001:**
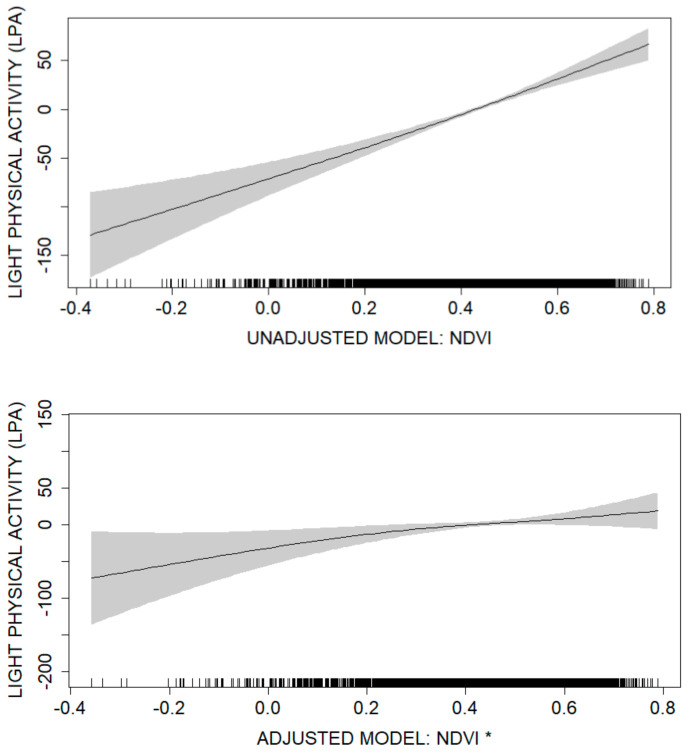
The association between residential greenness and the amount of light-intensity activity in the crude model and in the adjusted model. Values close to −1 indicate water bodies, rock, and snow. Values close to 0 (such as 0.2–0.3), in turn, indicate densely built surfaces or other surfaces with sparse vegetation. Highly positive values (>0.6) indicate areas with very dense and healthy green vegetation, such as forests and paddocks [41]. * The model was adjusted with: BMI kg/m^2^, physical strenuousness of work, children < 18 years old in the family, high education, daily alcohol intake (g), excellent/good perceived health and number of sport facilities.

**Table 1 ijerph-17-09202-t001:** Characteristics of the study population.

Characteristics	All (*n* = 5433)	Men (*n =* 3043)	Women (*n =* 2388)
NDVI, (SD) *	0.4268 (0.1486)	0.4315 (0.1481)	0.4231 (0.1489)
Weight, kg, (SD)	77.6 (16.1)	87.1 (14.8)	71.8 (14.7)
BMI, kg/m^2^, (SD)	26.8 (4.8)	27.3 (4.2)	26.4 (5.2)
Highly educated, *N* (%)	1443 (26.5)	538 (17.6)	905 (37.8)
Physical strenuousness of work, (SD) *	4 (3)	4 (3)	3 (2)
Children < 18 years old in the family (yes), *N* (%)	3500 (64.4)	1504 (49.4)	1996 (83.5)
Daily alcohol intake, g, (SD)	10.4 (16.9)	15.6 (21.8)	71.8 (14.7)
Smoking (yes), *N* (%)	1168 (21.4)	569 (18.6)	599 (25.0)
Total annual household income, €, (SD)	72,202 (298,532)	80,141 (439,388)	65,823 (74,487)
Excellent/good perceived health (yes), *N* (%)	3510 (64.6)	1485 (48.8)	2025 (84.7)
Living in semi-rural and rural areas (vs. urban area), *N* (%)	1891 (34.8)	867 (28.4)	1024 (42.8)
Number of sport facilities, (SD)	10 (14)	9 (13)	11 (14)

Note: Values are means if not otherwise stated. Values do not match due to missing values. * NDVI = values vary from −1 to +1. * Physically strenuous work = values vary from 1 to 9. SD, standard deviation.

**Table 2 ijerph-17-09202-t002:** Daily average MET-minutes (Metabolic Equivalent (SD)) of accelerometry measured PA according to socioeconomic, lifestyle habits, and residential environment.

Characteristics	LPA	MPA	VPA/VVPA	MVPA
All (*n* = 5433)	719 (193)	144 (88)	226 (160)	370 (209)
Men	697 (191)	188 (98)	234 (174)	422 (233)
Women	736 (193) **	109 (58) **	219 (147) *	328 (176) **
High education	662 (169) **	134 (71) **	236 (152) *	371 (188)
Low education	743 (198)	149 (93)	223 (163)	372 (217)
Children <18 years old (yes)	677 (196) **	146 (87) **	227 (154)	374 (203)
Children <18 years old (no)	733 (187)	136 (80)	225 (162)	362 (206)
Smoker	725 (188)	138 (87) **	330 (195) **	191 (143) **
Nonsmoker	725 (188)	150 (88)	378 (198)	227 (149)
Perceived health (excellent/good)	724 (190) *	147 (86) **	390 (203) **	243 (158) **
Perceived health (moderate/poor)	707 (200)	138 (90)	331 (214)	193 (159)
Living in semirural or rural areas	764 (196) **	152 (101) **	378 (237)	225 (175) *
Living in urban areas	695 (188)	139 (78)	225 (151)	365 (191)

Note. ** *p* < 0.001, * *p* < 0.05. *p*-values indicate statistical difference between groups.

**Table 3 ijerph-17-09202-t003:** Association between residential greenness (NDVI) and accelerometry measured LPA, MPA, VPA/VVPA, and MVPA (daily average minutes of MET-minutes) among middle-aged adults and men and women separately according to multivariable linear regression.

	All, *N* = 5433		Men, *N* = 2388		Women, *N* = 3040	
	Unadjusted B (95% CI)	Adjusted B (95% CI)	Unadjusted B (95% CI)	Adjusted B (95% CI)	Unadjusted B (95% CI)	Adjusted B (95% CI)
Light physical activity (LPA)						
NDVI (residential greenness)	174 (140, 209) **	70 (26, 114) **	224 (173, 275) **	140 (75, 204) **	142 (96, 188) **	17 (−42, 76)
Moderate physical activity (MPA)						
NDVI (residential greenness)	27 (11, 43) *	5 (−14, 24)	75 (48, 101) **	23 (−12, 60)	−22 (−36, −8) *	−12 (−32, 7)
Vigorous/very vigorous physical activity (VPA/VVPA)						
NDVI (residential greenness)	−20 (−48, 8)	−16 (−56, 23)	13 (−33, 61)	15 (−49, 80)	−49 (−84, −14) *	−36 (−86, 13)
Moderate–vigorous physical activity (MPA)						
NDVI (residential greenness)	7 (−29, 44)	−11 (−61, 39)	89 (25, 152) *	39 (−46, 125)	−71 (−113, −29) *	−48 (−108, 10)

Note. ** *p* < 0.001, * *p* < 0.05. *p*-values indicate statistical difference between groups.

## Data Availability

The datasets generated and/or analyzed during the study are available in the NFBC Project Center repository (https://www.oulu.fi/nfbc/node/47960).

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
