# Peer review of "Satellite Imaging-Based Residential Greenness and Accelerometry Measured Physical Activity at Midlife—Population-Based Northern Finland Birth Cohort 1966 Study"

_ijerph, 2020, doi:10.3390/ijerph17249202_

Round 1

Reviewer 1 Report

The MS is very interesting to urban forester and is good to read.

If authors can find the evidence to support the gender difference between NDVI and PA, it would be good paper.

Page 5
L 222
Table 1
Total annual ~, 80141 --> 80,141 , 65823 --> 65,823

L 273~275, double expression in backgrounds, please delete.
L 324 perhaps men prefer ~, men perhaps prefer
L 324 ~ 330
    <English expression is not enough to understand the gender difference in
      association between NDVI and LPA>
   # 54 reference seems to be an annual report of research organization.
  If you find the evidence to support the gender difference in cognition of NDVI and
  PA, the ms could be more developed. I would encourage you can find the
  evidence in cultural perspectives and historical backgrounds of leisures in Finland.

L 326 parental leave --> parental care (?), not clarified meaning

Did you check the association between the kinds of job and intensity of PA ?
PA would be related with the kinds of job of surveyors.

Author Response

Reviewer #1:

We appreciate the valuable comments given by the reviewer and we have made a detailed list of actions taken during the revision.

Comment: The MS is very interesting to urban forester and is good to read.

Comment: If authors can find the evidence to support the gender difference between NDVI and PA, it would be good paper.

Response: The reasons underlying our finding can only be speculated. We went through   literature and scientific evidence and found no more supportive studies. Thus, our finding is novel and more future are needed to confirm this phenomenon.

Comment:

Page 5
L 222
Table 1

Total annual ~, 80141 --> 80,141 , 65823 --> 65,823

Response: The numbers were modified as requested.

Comment: L 273~275, double expression in backgrounds, please delete.

Response: We apologize but we do not fully understand this comment. The sentence in lines (page 8, lines 275 – 277) is as follows : “The purpose of this study was to investigate the relationship between satellite imaging–based residential greenness and the amount of accelerometry-measured PA at different intensities, with a special emphasis on LPA and gender differences.” We do not see any double expression in this sentence, please clarify. We are of course prepared to clarify the sentence if needed.

Comment: L 324 perhaps men prefer ~, men perhaps prefer

Response: The expression was modified.

Comment: L 324 ~ 330
    <English expression is not enough to understand the gender difference in association between NDVI and LPA>

Response: We have clarified the text (page 9, lines 324 – 329). The gender difference in association between NDVI and LPA can only be speculated. According to the Working Paper of the Finnish Forest Research Institute [55] women spend more time outdoors within their residential areas such as their own yards, parks and natural environments. This may have covered the association between greenness and women’s LPA. However, more scientific evidence is needed to confirm our finding.

Comment: # 54 reference seems to be an annual report of research organization.

Response: We agree with the reviewer that the reference is not a scientific report. However, it not an annual report either but a comprehensive report on the outdoor activities of Finnish women and men and supports our speculation. We have now modified the sentence including the reference 54 (due to the changes the number of the reference is 55)

Comment: If you find the evidence to support the gender difference in cognition of NDVI and PA, the ms could be more developed. I would encourage you can find the evidence in cultural perspectives and historical backgrounds of leisures in Finland.

Response: We thank the reviewer for this comment, but unfortunately, we found no more scientific evidence supporting our finding. Hopefully, in future, in the Northern Finland Birth Cohort studies, we’ll be able to further study this phenomenon also.

Comment: L 326 parental leave --> parental care (?), not clarified meaning

Response: The whole paragraph was modified according to the comments of the reviewer and this sentence removed.

Comment: Did you check the association between the kinds of job and intensity of PA? PA would be related with the kinds of job of surveyors.

Response: The kind of job is important regarding our research questions because different kind of jobs have different demands and strenuousness. Thus, we analyzed the association between physical strenuousness of work and different intensities of physical activity. Due to the strong association, it was used as a covariate in the models.

Reviewer 2 Report

The manuscript explored the association between neighborhood greenness and residents’ physical activity intensity, which can add some new knowledge to current literature.

p.107 Please introduces more about the reliability and validity of the Polar Active in measuring physical activity intensity. Has it been widely used?

  1. 238. Have the user used the lineary regression or logistd regression ?

I suggest the author provided more pictures about the sites and how to calculate the amount of greenness.

Author Response

Reviewer #2:

We appreciate the valuable comments given by the reviewer and we have made a detailed list of actions taken during the revision.

Comment: The manuscript explored the association between neighborhood greenness and residents’ physical activity intensity, which can add some new knowledge to current literature.

Comment: p.107 Please introduces more about the reliability and validity of the Polar Active in measuring physical activity intensity. Has it been widely used?

Response: Polar Active measuring has been used in many studies in Finland. and it has shown to be reliability in measuring energy expenditure [32].

Comment: 238. Have the user used the linear regression or logistic regression?

Response: We have used linear regression and it has been mentioned in the text (page 5, line 203)

Comment: I suggest the author provided more pictures about the sites and how to calculate the amount of greenness.

Response: The principles of the tool NDVI which represents greenness has been described in the text (page 4, lines 150 – 151). We also gave examples of how the values of the index can be interpret. Therefore, we have not added detailed pictures to the text. Even though pictures are good in visualization the results, we think in this case, pictures would have been too coarse to bring any additional information to the study. However, if needed, we can provide some figures to be included e.g. in a supplement file.

Reviewer 3 Report

The authors aim to find the quantitative relationships between the human activities and the amount of the greenery by a large amount of data and objective measurement. The research has sufficient data support, specific and complete experiments, clear overview and sufficient discussion. I believe the research provided important evidence for studies about the relation between greenery and human physical health. The authors should consider the following revisions.

  • Abstract:

It is recommended that you put “aims” before “methods”.

  • Background:

It will be more complete if you add the overview: why the methods to be used can solve the current problems. These overview words should probably appear after the third paragraph.

  • Methods:

Line 128-129, references required.

Line 131, “1km”. It is recommended to consider different buffer sizes in the future study or the revised version. Although there were studies that used 1 km as the parameter, still many other studies used other buffer sizes, like 500m, 3km, etc.

  • Result: In the firth row of Table 1, the sum of “men” and “women” is not equal to “all”.

Results by “adjusted” and “unadjusted” beta distribution appears frequently in this manuscript. Please further explain the meaning and the differences between “adjusted” and “unadjusted” model. Which model is recommended?

  • Discussion: In personal opinion, it is not always necessary to find enough reasons for all the experimental results. If there is not enough evidence to support a possible point of view, I advise against guessing too much. For example, Line 321-331, reduce speculation appropriately or try to find a stronger evidence.
  • It is recommended to present regional limitations and age limitations in “Strengths and weaknesses”, and even in the conclusions, to make the article more rigorous.

Author Response

Reviewer 3.

We appreciate the valuable comments given by the reviewer and we have made a detailed list of actions taken during the revision.

Comment: The authors aim to find the quantitative relationships between the human activities and the amount of the greenery by a large amount of data and objective measurement. The research has sufficient data support, specific and complete experiments, clear overview and sufficient discussion. I believe the research provided important evidence for studies about the relation between greenery and human physical health. The authors should consider the following revisions.

Abstract:

Comment: It is recommended that you put “aims” before “methods”.

Response: The structure of the abstract was modified according to the comment.

Background:

Comment: It will be more complete if you add the overview: why the methods to be used can solve the current problems. These overview words should probably appear after the third paragraph.

Response: We have now added description of the NDVI methodology and its usability to the Background section (page 2, lines 82 – 85).

Comment: Line 128-129, references required.

Response: Geographic Information Systems was generally described in the sentence and it is a widely used method. Therefore, we assume reference is not needed.

Comment: Line 131, “1km”. It is recommended to consider different buffer sizes in the future study or the revised version. Although there were studies that used 1 km as the parameter, still many other studies used other buffer sizes, like 500m, 3km, etc.

Response: The selected buffer size in this study was based on previous studies (page 3, lines 131 – 132) and our own experience on different population studies among age groups. In addition, 1km buffer has been considered suitable when studying physical activity (especially light physical activity) that is performed within residential environment.

Comment: Result: In the firth row of Table 1, the sum of “men” and “women” is not equal to “all”.

 Response: The numbers do not match due to missing values. This was added below the table 1.

Comment: Results by “adjusted” and “unadjusted” beta distribution appears frequently in this manuscript. Please further explain the meaning and the differences between “adjusted” and “unadjusted” model. Which model is recommended?

Response: The unadjusted (crude) betas represent the original observations as such. However, they are influenced by a variety of personal and environmental factors, while our study focused on the effect of greenness. The unadjusted results are valuable as such, but they may hide the effect of greenness, the focus of the study. Therefore, the results were adjusted for potential confounders (having hildren < 18 in the family, gender, level of education, physically strenuous work, number of sport facilities, BMI (kg/m2), daily alcohol intake, good self-rated health). The adjusted results show the independent effect or greenness. We actually found that many crude associations disappeared when adjustments were made. Thus, the crude associations were partly explained by the confounding factors mentioned above. The crude and unadjusted regression coefficients both tell their own story – both are needed.

Comment: Discussion: In personal opinion, it is not always necessary to find enough reasons for all the experimental results. If there is not enough evidence to support a possible point of view, I advise against guessing too much. For example, Line 321-331, reduce speculation appropriately or try to find a stronger evidence.

Response: That is an important point and we totally agree with you. We have clarified the Discussion section according to the comments of the reviewer (page 9, lines 324 – 329).

Comment: It is recommended to present regional limitations and age limitations in “Strengths and weaknesses”, and even in the conclusions, to make the article more rigorous.

Response: We have modified the section “Strengths and weaknesses” (page 10, lines 362 - 364).